# On a Class of Positive Definite Operators and Their Application in Fractional Calculus

Temirkhan Aleroev 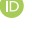

The National Research Moscow State University of Civil Engineering (NRU MGSU), 129337 Moscow, Russia; aleroev@mail.ru

**Abstract:** This paper is devoted to the spectral analysis of one class of integral operators, associated with the boundary value problems for differential equations of fractional order. Approximation matrices are also investigated. In particular, the positive definiteness of the studied operators is shown, which makes it possible to select areas in the complex plane where there are no eigenvalues of these operators.

**Keywords:** persymmetric matrix; eigenvalues; fractional derivative; positive definiteness

## 1. Introduction

As is known in fractional calculus and in the theory of mixed-type equations, an important role is played by the potential function

$$\frac{1}{2}\int\limits_0^1 \frac{u(t)dt}{|x-t|^{1/\rho}}$$

with density $u(t)$ and with a power kernel $\frac{1}{|x-t|^{1/\rho}}$, which is positive definite for $0 < \frac{1}{\rho} < 1$ according to a fact established by Tricomi [1–8]. There are papers in which various generalizations of this result were given. First of all, we should note the paper of Gellerstedt [9], where an operator of the following form was investigated for positive definiteness

$$P_{01}^{\varphi}u(x) = \int\limits_0^1 \varphi(|x-t|)u(t)dt$$

where

$$\varphi(|x-t|) = |x-t|^{m/(m+2)}P_0(c|x-t|^{4/(m+2)})$$

([9], page 41) which is some generalization of the operator

$$\frac{1}{2}\int\limits_0^1 \frac{u(t)dt}{|x-t|^{1/\rho}}.$$

Another direction was started in [10,11] where, in particular, it was shown that the operator $\widetilde{A}_\rho : L_2 \to L_2$, for $0 < \frac{1}{\rho} < 1$ (where $\widetilde{A}_\rho = \int\limits_0^x (x-t)^{1/\rho-1}u(t)dt$) is sectorial and also that the values of the form $(\widetilde{A}_\rho u, u)$, for $1 < \frac{1}{\rho} < \infty$, fill the whole complex plane [10]. This manuscript is devoted to the study of the positive definiteness of operators of the form

$$A_\gamma^{[\alpha,\beta]}u(x) = c_\alpha \int\limits_0^x (x-t)^{1/\alpha-1}u(t)dt + c_{\alpha,\beta}\int\limits_0^1 x^{1/\rho-1}(1-t)^{1/\gamma-1}u(t)dt,$$

which are finite-dimensional perturbations (finite-dimensional perturbations of a special kind) of a fractional integration operator of a special kind.

We suggest a principally new wide class of positive definite operators, which play an important role in fractional calculus and in applications. The obtained results are used to study some very important properties of functions of the Mittag-Leffler type.

## 2. On the Positive Definiteness of Operators of the Kind $A_\gamma^{[\alpha,\beta]}$

Let us consider the operator

$$A_\gamma^{[\alpha,\beta]}u(x) = c_\alpha \int\limits_0^x (x-t)^{\frac{1}{\alpha}-1}u(t)dt + c_{\beta,\gamma} \int\limits_0^1 x^{\frac{1}{\alpha}-1}(1-t)^{\frac{1}{\alpha}-1}u(t)dt.$$

This operator arises in the process of determining the solutions of boundary value problems for fractional differential equations [12].

Let us show that this operator (for specific $\alpha, \gamma, \rho$) is positive definite. To highlight the main ideas, let us consider the simplest case. Let us consider in space $L_2(0,1)$ the operator $A_\gamma^{[\alpha,\beta]}$ for $\alpha = \beta = \gamma = \rho$, $0 < \rho < 2$, i.e., we consider the operator $A_\rho$ [13]. The more important case is for $0 < \rho < 1$, as in this case, the operator $A_\rho$ corresponds to the differential equations of order more than 1. The case for $1 < \rho < \infty$ in fractional calculus is not so interesting but to complete our investigation, we will to consider some results for this case too.

First of all we, note that the first term of operator $A_\rho$ is a fractional integral $J^{1/\rho}$ of order $1/\rho$.

Let us designate

$$\widetilde{A}_\rho u = \int\limits_0^x (x-t)^{1/\rho-1}u(t)dt.$$

Obviously, the operator $\widetilde{A}_\rho$ is different from the operator $J^{1/\rho}$ by the positive constant. However, in the following, for easy reading, we will use the operator $\widetilde{A}_\rho$ and do not pay attention to this difference. As is known in fractional calculus and in the theory of mixed-type equations, an important role is played by the potential

$$\frac{1}{2}\int\limits_0^1 \frac{u(t)dt}{|x-t|^{(1/\rho)}} \tag{1}$$

with density $u(t)$ and with a power kernel $\frac{1}{|x-t|^{(1/\rho)}}$ which is positive definite, for $0 < 1/\rho < 1$, and this fact was established by F. Tricomi [8].

F. Tricomi [8] showed that the symmetric component of the operator $A_\rho$, i.e.,

$$\widetilde{A}_{\rho_R}u = \frac{1}{2}\int\limits_0^1 \frac{u(t)dt}{|x-t|^{1/\rho}}$$

is fixed-sign, i.e.,

$$(\widetilde{A}_{\rho_R}u, u) = \frac{1}{2}\int\limits_0^1\int\limits_0^1 \frac{u(t)\overline{u}(x)dtdx}{|x-t|^{1/\rho}} \geq 0$$

is positive definite, i.e., $(\widetilde{A}_{\rho_R}u, u) \geq 0$. It should be noted that the operator $\widetilde{A}_\rho$ is strictly definite $(\widetilde{A}_\rho u, u) > 0$ (the equality sign holds if and only if $u = 0$). A little later, Matsaev and Palant [11] showed that the operator $\widetilde{A}_\rho$ is sectorial $(0 < 1/\rho < 1)$, that is, the values of the form $(\widetilde{A}_\rho u, u)$ lie in the angle

$$|arg\lambda| < \frac{\pi}{2\rho}.$$

Further, Gokhberg and Krein [10] showed that the values of the form $(\widetilde{A}_\rho u, u)$ for $(1 < 1/\rho < \infty)$ fill the whole complex plane. This paper provides further analysis of these operators.

There is another well-known result. The operator $A_\rho$, for $1 < \rho < 2$ for $(A_\rho : L_2 \to L_2)$, is positive definite.

Let us now formulate and prove the following theorem

**Theorem 1.** *Operator* $-A_\rho$ *is positive definite for* $1/2 < 1/\rho < 1$ *and* $(A_\rho : L_2 \to L_2)$.

**Proof.** It is known that the operator $A$ is called positive definite if $(Au, u) > 0$, $(u \neq 0)$. However, it is very difficult to verify this condition directly. Therefore, we will use the matrix approximation of the operator $A_\rho$ [6]. As in [6], we denote the corresponding matrix by $T_{n-1}(\mu)$, $\mu = \frac{1}{\rho} - 1$ $(1 < \mu < 2)$

$$T_{n-1}(\mu) = \begin{pmatrix} (\frac{1}{n})^\mu(\frac{n-1}{n})^\mu & (\frac{1}{n})^\mu(\frac{n-2}{n})^\mu & \cdots & (\frac{1}{n})^\mu(\frac{1}{n})^\mu \\ (\frac{2}{n})^\mu(\frac{n-1}{n})^\mu - (\frac{1}{n})^\mu & (\frac{2}{n})^\mu(\frac{n-2}{n})^\mu & \cdots & (\frac{2}{n})^\mu(\frac{1}{n})^\mu \\ \vdots & \vdots & \ddots & \vdots \\ (\frac{n-1}{n})^\mu(\frac{n-1}{n})^\mu - (\frac{n-1}{n})^\mu & (\frac{n-1}{n})^\mu(\frac{n-2}{n})^\mu - (\frac{n-3}{n})^\mu & \cdots & (\frac{n-1}{n})^\mu(\frac{1}{n})^\mu \end{pmatrix}.$$

The system of eigenfunctions of the $K$ is complete in the domain of values of the integral operator $Kf$ if and only if both sides in this inequality are equal.

It is known that the operator $A$ is called positive definite if $(Au, u) > 0$, $(u \neq 0)$. However, it is very difficult to verify this condition directly. Therefore, we will use the matrix approximation of the operator $A_\rho$ [6]. As in [6], we denote the corresponding matrix by $T_{n-1}(\mu)$, $\mu = \frac{1}{\rho} - 1$

$$T_{n-1}(\mu) = \begin{pmatrix} (\frac{1}{n})^\mu(\frac{n-1}{n})^\mu & (\frac{1}{n})^\mu(\frac{n-2}{n})^\mu & \cdots & (\frac{1}{n})^\mu(\frac{1}{n})^\mu \\ (\frac{2}{n})^\mu(\frac{n-1}{n})^\mu - (\frac{1}{n})^\mu & (\frac{2}{n})^\mu(\frac{n-2}{n})^\mu & \cdots & (\frac{2}{n})^\mu(\frac{1}{n})^\mu \\ \vdots & \vdots & \ddots & \vdots \\ (\frac{n-1}{n})^\mu(\frac{n-1}{n})^\mu - (\frac{n-1}{n})^\mu & (\frac{n-1}{n})^\mu(\frac{n-2}{n})^\mu - (\frac{n-3}{n})^\mu & \cdots & (\frac{n-1}{n})^\mu(\frac{1}{n})^\mu \end{pmatrix}.$$

$\square$

The matrix $T_{n-1}(\mu)$ has many useful properties. In particular, this matrix is positive, persymmetric, indecomposable, etc. It is known [6] that one of the necessary conditions for the positive definiteness of a matrix is the positivity of all its lead main minors. The fact that these minors are positive was shown in [6]. Next, we need the following lemmas.

**Lemma 1.** *The minors*

$$A\begin{pmatrix} i_1 & i_2 & \cdots & i_r \\ j_1 & j_2 & \cdots & j_r \end{pmatrix}$$

*of the matrix (here $\mu > 1$)*

$$T_{n-1}(\mu) = \begin{pmatrix} (\frac{1}{n})^\mu(\frac{n-1}{n})^\mu & (\frac{1}{n})^\mu(\frac{n-2}{n})^\mu & \cdots & (\frac{1}{n})^\mu(\frac{1}{n})^\mu \\ (\frac{2}{n})^\mu(\frac{n-1}{n})^\mu - (\frac{1}{n})^\mu & (\frac{2}{n})^\mu(\frac{n-2}{n})^\mu & \cdots & (\frac{2}{n})^\mu(\frac{1}{n})^\mu \\ \vdots & \vdots & \ddots & \vdots \\ (\frac{n-1}{n})^\mu(\frac{n-1}{n})^\mu - (\frac{n-1}{n})^\mu & (\frac{n-1}{n})^\mu(\frac{n-2}{n})^\mu - (\frac{n-3}{n})^\mu & \cdots & (\frac{n-1}{n})^\mu(\frac{1}{n})^\mu \end{pmatrix}.$$

*for $i_k \leq j_k$, $1 \leq k \leq r$, are positive. Moreover, they are equal to*

$$(n^\mu)^{r-1}(n-j_r)^\mu (i_1)^\mu b_{r(r-1)} b_{(r-1)(r-2)} \ldots b_{21},$$

*where*

$$b_{ki} = \begin{cases} (i_k - j_i)^\mu, & i_k > j_i \\ 0, & i_k < j_i \end{cases}$$

**Proof.** Let us consider the minor

$$M_r = A \begin{pmatrix} i_1 & i_2 & \cdots & i_r \\ j_1 & j_2 & \cdots & j_r \end{pmatrix}.$$

For $i_k \leq j_k$, $1 \leq k \geq r$, we may represent $M_r$ as follows

$$M_r = \begin{pmatrix} i_1^\mu \\ i_2^\mu \\ i_3^\mu \\ \vdots \\ i_{r1}^\mu \end{pmatrix} \left( (n-j_1)^\mu (n-j_2)^\mu \ldots (n-j_r)^\mu \right) - n^\mu \begin{pmatrix} 0 & 0 & 0 & \cdots & 0 \\ b_{21} & 0 & 0 & \cdots & 0 \\ \vdots & \vdots & \vdots & \ddots & \vdots \\ b_{r1} & b_{r2} & b_{r3} & \cdots & b_{r(n-1)} \end{pmatrix}$$

for

$$b_{ki} = \begin{cases} (i_k - j_i)^\mu, & i_k > j_i \\ 0, & i_k < j_i \end{cases}$$

To calculate the determinant $M_r$, we consider

$$\det(M_r - \lambda I) = (-1)^r \det(\lambda I - M_r) = (-1)^r \det(\widetilde{A} - xy^T) =$$
$$(-1)^r \lambda^r (1 - y^T \widetilde{A}^{-1} x).$$

Here,

$$\widetilde{A} = n^\mu \begin{pmatrix} \frac{\lambda}{n^\mu} & 0 & 0 & \cdots & 0 \\ b_{21} & \frac{\lambda}{n^\mu} & 0 & \cdots & 0 \\ \vdots & \vdots & \vdots & \ddots & \vdots \\ b_{r1} & b_{r2} & b_{r3} & \cdots & \frac{\lambda}{n^\mu} \end{pmatrix},$$

$$x = \begin{pmatrix} i_1^\mu \\ i_2^\mu \\ i_3^\mu \\ \vdots \\ i_{r1}^\mu \end{pmatrix}, y^T = \left( (n-j_1)^\mu (n-j_2)^\mu \ldots (n-j_r)^\mu \right).$$

It is clear that

$$x_{r1} = \left(-\frac{n^\mu}{\lambda}\right)^{r-1} b_{r(r-1)} b_{(r-1)(r-2)} \ldots b_{21} x_1 \ldots$$

$$= (-1)^{r-1} \left(\frac{n^\mu}{\lambda}\right)^{r-1} b_{r(r-1)} b_{(r-1)(r-2)} \ldots b_{21} + \ldots$$

So,

$$\det(M_r - \lambda I) = (-1)^r \lambda^r (1 - y^T \widetilde{A}^{-1} x) =$$
$$(-1)^r \lambda_r \left(1 - (n-j_r)^\mu i_1^r x_{r1} + \ldots\right)$$

from this follows

$$\det(M_r) = (n^\mu)^{r-1}(n - j_r)^\mu (i_1)^\mu b_{r(r-1)} b_{(r-1)(r-2)} \ldots b_{21}$$

which proves the lemma. $\square$

To prove that the matrix $T_{n-1}(\mu)$ is positive definite, we have studied the real component of this matrix

$$T_R = \frac{1}{2}(T_{n-1}(\mu) + T_{n-1}^*(\mu)).$$

Using the high-level mathematical package MATLAB, the eigenvalues of the matrix $T_R$ were considered for various values of $\mu$ and the dimension of the matrix $N$. It was shown that all eigenvalues of the matrix $T_R$, for any $N \le 3000$ and $\mu > 0$, are positive, that is, the above calculations confirm the hypothesis that the matrix $T_R$ is positive definite. This became the basis for us to assume that the matrix $T_{(n-1)}(\mu)$ under study is positive definite. It is natural that the operator $A_\rho$ corresponding to the matrix $T_{(n-1)}(\mu)$ will also be positive definite.

We give a strong proof of the positive definiteness of the matrix $T_R(\mu)$. First, let us write the matrices $T_6(1/2)$, $T_6^*(1/2)$, $T_R(1/2)$ using the MATLAB package

$$T_6(1/2) = \begin{pmatrix} 2.4495 & 2.2361 & 2 & 1.7321 & 1.4142 & 1 \\ 0.8184 & 3.1623 & 2.8284 & 2.4495 & 2 & 1.4142 \\ 0.5010 & 1.2272 & 3.4641 & 3 & 2.4495 & 1.7321 \\ 0.3164 & 0.7305 & 1.3542 & 3.4641 & 2.8284 & 2 \\ 0.1857 & 0.4174 & 0.7305 & 1.2272 & 3.1623 & 2.2361 \\ 0.0839 & 0.1857 & 0.3164 & 0.5010 & 0.8184 & 2.4495 \end{pmatrix}$$

$$T_6^*(1/2) = \begin{pmatrix} 2.4495 & 0.8184 & 0.5010 & 0.3164 & 0.1857 & 0.0839 \\ 2.2361 & 3.1623 & 1.2272 & 0.7305 & 0.4174 & 0.1857 \\ 2 & 2.8284 & 3.4641 & 1.3542 & 0.7305 & 0.3164 \\ 1.7321 & 2.4495 & 3 & 3.4641 & 1.2272 & 0.5010 \\ 1.4142 & 2 & 2.4495 & 2.8284 & 3.1623 & 0.8184 \\ 1 & 1.4142 & 1.7321 & 2 & 2.2361 & 2.4495 \end{pmatrix}$$

$$T_R(1/2) = \begin{pmatrix} 2.4495 & 1.5272 & 1.2505 & 1.0242 & 0.8 & 0.5420 \\ 1.5272 & 3.1623 & 2.0278 & 1.5900 & 1.2087 & 0.8 \\ 1.2505 & 2.0278 & 3.4641 & 2.1771 & 1.59 & 1.0242 \\ 1.0242 & 1.5900 & 2.1771 & 3.4641 & 2.0278 & 1.2505 \\ 0.8 & 1.2087 & 1.59 & 2.0278 & 3.1623 & 1.5272 \\ 0.5420 & 0.8 & 1.0242 & 1.5272 & 1.5272 & 2.4495 \end{pmatrix}$$

That is, the following statements hold.

**Lemma 2.** *For any $i_0 \le j$, and $\mu > 1$, the following relations hold*

$$a_{i_0,j} \ge a_{i_0,j+1}, \ i_0 \le j;$$

$$a_{i_0,j} < a_{i_0,j+1}, \ i_0 > j.$$

**Proof.** We write the formula for the general element of the matrix

$$a_{ij} = (Ni - ij)^\mu - \theta(i,j)(Ni - Nj)^\mu.$$

Obviously, the elements under the main diagonal are calculated as follows, $\mu > 1$

$$a_{ij} = (Ni - ij)^\mu - (Ni - Nj)^\mu, \ i > j,$$

and the elements under the main diagonal are

$$a_{ij} = (Ni - ij)^\mu, \ i > j.$$

So, these formulas show that the elements located above the main diagonal decrease. To consider the elements under the main diagonal, we introduce the generating function

$$\varphi(x) = (Ni - ix)^\mu - \theta(i,j)(Ni - Nx)^\mu, \ \mu \in (1,2), \ x \in [1, N].$$

Obviously, the derivative of this function is positive on the segment $x \in [1, N]$, which means that the function $\varphi(x)$ increases on the segment $x \in [1, N]$. This completes the proof. $\square$

**Lemma 3.** *For any fixed $j_0 < i$, the following relations*

$$a_{i,j_0} \geq a_{i+1,j_0}, \ i \geq j_0;$$

$$a_{i,j_0} < a_{i+1,j_0}, \ i < j_0$$

*hold.*

**Proof.** The proof of Lemma 2 is similar to the proof of Lemma 1. $\square$

**Lemma 4.** *The statements of Lemmas 1 and 2 are valid for the matrices $T_n^T(\mu)$ ($T_n^T(\mu)$ is the transposed matrix, and it shall be reminded that $T(\mu)$ for $\mu = 1$ was studied by Krein [10], for $\mu = 2$, it was studied in [6]).*

**Lemma 5.** *The statements of Lemmas 1–3 are also valid for the matrices $T_R(\mu) = \frac{T_n(\mu) + T_n^T(\mu)}{2}$.*

Using these lemmas, we prove the following theorem.

**Lemma 6.** *The matrix $T_R(\mu) = \frac{T_n(\mu) + T_n^T(\mu)}{2}$ is positive definite for $\mu \in (1,2)$.*

**Proof.** It is obvious that all main lead minors of the matrix $T_R(0)$ are non-negative. In the same way, all main lead minors of the matrix $T_R(1)$ are positive.

Let us show that for $\mu \in (1,2)$, all main lead minors of the matrix are $T_R(\mu) \neq 0$. To do this, it is enough to prove that all the rows (columns) of the leading main lead minors of the matrix are linearly independent. In proving this statement, without loss of generality, for definiteness, we consider rows with numbers $k$ and $k + 1$. Then, it suffices to note that, by Lemma 4, that $\frac{a_{k,1}}{a_{k+1,1}} < 1$ and $\frac{a_{k,k+1}}{a_{k,k}} > 1$, which proves the linear independence of these rows.

Let us introduce the following function

$$det T_R(\mu) = \Delta(\mu), \ \mu \in (1,2).$$

It is known that $\Delta(2) > 0$ and $\Delta(1) > 0$.

From the above-provided statements follows that the matrix $T_{n-1}(\mu)$ is positive definite. So, the operator $A_\rho$ is positive definite too, and this proves the lemma. $\square$

From this very important theorem, it follows that the operator $A_\rho$ is positive definite for $1/2 < \rho < \infty$.

## 3. Conclusions

So, our spectral analysis of the operators generated by boundary value problems for fractional differential equations and boundary conditions of Sturm–Liouville type, using

matrix calculus, shows that the spectral structure of these operators can be studied by the matrices we studied above.

## 4. Discussion

Operators generated (induced) by a differential expression of a fractional order and boundary conditions of the Sturm–Liouville type are non-self-adjoint and their spectral structure is almost not studied. The methods proposed by the authors are fundamentally new. They allow to study the completeness of systems of eigenfunctions and associated functions of these operators.

**Author Contributions:** All results of this manuscript were provided by T.A. The author have read and agreed to the published version of the manuscript.

**Funding:** This research received no external funding.

**Institutional Review Board Statement:** Not applicable.

**Informed Consent Statement:** Not applicable.

**Data Availability Statement:** Not applicable.

**Conflicts of Interest:** The author declares no conflict of interest.

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
