# Peer review of "On a Class of Positive Definite Operators and Their Application in Fractional Calculus"

_axioms, doi:10.3390/axioms11060272_

Round 1
Reviewer 1 Report
This paper is devoted to the spectral analysis of one class of integral operators, associated to the boundary value problems for differential equations of fractional order. In particular, it was shown the positive definiteness of studying operators, which makes it possible to select areas in the complex plane without eigenvalues of of these operators. The methods proposed by the author seem to be new. They allow the study of the completeness of systems of eigenfunctions and the associated functions of these operators.
However, the English language of the article need to be checked. For example in the abstract it was written "In particular, was shown " with subjective missed which was repeated throughout the article.
Another example, In page 6 line 84 it was written " Using these lemmas we prove the following theorem"
then it was written a Lemma instead of Theorem.
Moreover, in the conclusion and the discussion he used the plural rather than the singular, referring to the author.
Author Response
Dear reviewer! Thank you very much for your comments, there were taken into account in manuscript.
Reviewer 2 Report
I have read this paper very carefully and found it not very interesting for the scientific readers. But It can be accepted after the following revision.
1) Author must reduce the plagiarism, It's about more than 40%.
2) Author must improve the introduction section by providing the latest references in this field.
3) At page 2, line 22, "Aρ is fractional integral J^1/ρ of order 1/ρ", author must provide that which kind of fractional integral?
4) The main theorem of this manuscript is Theorem 1 on page 3, in the proof of this theorem, the line number 33-37 and 38-42 are same (duplication) author must check it.
5) Author must provide any example which support this study
Author Response
Dear reviewer! Thank you very much for your comments
- The plagiarism was reduced.
- The introduction was improved
- Here J^1/ρ is an operator of fractional integration of order 1/ρ in Riemann-Liouville sense
- These was checked lines
- For 1/ρ =2 we obtain the positive defined operator, corresponding to the boundary value problem
Reviewer 3 Report
See enclosed report

Author Response

(The authors gave the same response as above.)
